

# A fixed agreement—consequences of brood size manipulation on alternation in blue tits

Maaike Griffioen, Wendt Müller and Arne Iserbyt

Department of Biology, Behavioural Ecology and Ecophysiology Research Group, University of Antwerp, Wilrijk, Belgium

## ABSTRACT

Recent studies have proposed that conditional cooperation may resolve sexual conflict over the amount of care provided by each parent. Such conditional cooperation may allow parents to equalize their investment by alternating their provisioning visits. This alternated pattern of male and female visits, that is, alternation, is thought to stimulate each other's investment leading to higher levels of provisioning and potential benefits for offspring development. However, experimental studies testing the role of alternation as an adaptive parental strategy to negotiate the level of investment are still absent. Therefore, we manipulated blue tit (*Cyanistes caeruleus*) parents by temporarily changing their brood sizes to induce changes in demand and thus visit rates. Parents were expected to visit more— assuming that prey sizes were constant—and alternate at higher levels when confronted with an enlarged brood given the greater potential for sexual conflict. In contrast, in reduced broods visit rates and alternation may become lower due to the smaller investment that is needed for reduced broods. We show that the level of alternation did not differ in response to the manipulated brood sizes, despite a directional change in visit rates for enlarged and reduced broods as expected. Nestlings did not benefit from high levels of alternation as no effects on nestling mass gain were present in either of the different manipulations. These findings indicate that alternation does not serve as a mechanism to motivate each other to feed at higher rates. Parents hence appeared to be inflexible in their level of alternation. We therefore suggest that the level of alternation might reflect a fixed agreement about the relative investment by each of the caring parents.

## INTRODUCTION

Biparental care is relatively rare among species in the animal kingdom, except among bird species, with more than 81% of the species showing this kind of care (*Cockburn, 2006*). To provide biparental care, two unrelated individuals get together in order to successfully raise their offspring. It secures a joined fitness benefit via improved offspring development and survival, however, both parents have to individually pay for the costs of providing care (*Trivers, 1972*). Therefore, a conflict is expected between parents about the amount

Corresponding author
Maaike Griffioen,
maaike.griffioen@uantwerpen.be

of investment that each should make to their current brood. It is in both parents' interest that their partner provides more care during the breeding attempt so that they themselves can retain energy for self-maintenance and future reproduction (*Stearns, 1989*). How parents resolve this conflict and how this inflicts their offspring has been given much attention since the formulation of the sexual conflict theory by *Trivers (1972)*. One of the first models hypothesized that each parent could only make a single independent investment so that individual investment could only change on an evolutionary time scale (*Houston & Davies, 1985*). Whereas more recent models implemented the possibility that parents are able to adjust their investment in response to their partner on a behavioral time scale (*Mcnamara, Gasson & Houston, 1999*; *Johnstone & Hinde, 2006*; *Lessells & McNamara, 2012*), which better captures the natural situation. However, both models predict that parents should invest at suboptimal levels of care, which may reduce offspring fitness but avoids exploitation by the partner (*Mcnamara, Gasson & Houston, 1999*; *McNamara et al., 2003*; *Johnstone & Hinde, 2006*; *Lessells & McNamara, 2012*; *Johnstone et al., 2014*). Recently, a more cooperative strategy has been proposed for the resolution of sexual conflict, namely conditional cooperation, which may avoid or reduce such costs of negotiation (*Johnstone et al., 2014*).

In humans, conditional cooperation implies that individuals are more likely to contribute to a public good when others are also willing to do so, which may lead to more efficient, higher overall levels of investment (*Gächter, 2007*; *Johnstone et al., 2014*). In the context of biparental care, conditional cooperation could be defined as the tendency of a parent to invest more when the partner invests as well. In other words, parents may become more motivated to feed after their partner has fed and therefore will speed up their visit rate. This may result in the pattern of alternated feeding visits, as well as greater and more equal levels of investment (*Johnstone et al., 2014*). Some recent studies indeed provided empirical support showing that parents alternated their visits more than predicted by chance (great tits *Parus major*: *Johnstone et al., 2014*; long-tailed tits *Aegithalos caudatus*: *Bebbington & Hatchwell, 2016*; canaries *Serinus canaria*: *Iserbyt et al., 2017*; chestnut-crowned babblers *Pomatostomus ruficeps*: *Savage et al., 2017*) and that higher total visit rates relate to higher levels of alternation (*Bebbington & Hatchwell, 2016*). Consequently, this raises the question whether higher levels of alternation also benefit offspring development. Evidence is still limited and confined to unmanipulated conditions, while further experimental studies are needed to answer the questions whether alternation forms a parental strategy to negotiate about investment and how this affects the offspring (*Iserbyt et al., 2019*).

The aim of the study was therefore to test whether the pattern of alternated male and female provisioning visits is maintained when the nest conditions change, and whether higher levels of alternation benefit the offspring. To this end, we performed a temporary brood size manipulation experiment in blue tit (*Cyanistes caeruleus*) parents in which brood size was either increased or decreased (*Parejo & Danchin, 2006*; *García-Navas & Sanz, 2010*). Previous studies have shown that brood size manipulations affect parental visit rates (see review in *Gow & Wiebe, 2014*). When brood size is increased, parents are expected to increase their provisioning rate to meet the elevated brood demand,
resulting in higher investment costs to both parents. Such elevated demand may therefore intensify the sexual conflict, which may be ameliorated via conditional cooperation through increasing individual costs of parents. When alternation improves between pair members, the likelihood of exploitation may be reduced because parents motivate each other to visit at higher rates. Thus alternation and visit rates are expected to rise in the situation that brood size is increased, assuming that prey size remains constant. When brood size is decreased, parents will likely reduce their alternation level and visit rate because of the lower sexual conflict over parental care. Finally, when alternation represents a form of conditional cooperation, in which parents enhance each other's willingness to invest and increase their visit rates accordingly, then it should benefit the offspring. However, the offspring can only benefit from a high alternation when the profitability of the delivered prey items remain unaltered. If parents cheat in delivered prey sizes, then parents cannot rely on their partners' visit rate as an honest signal of parental investment.

## MATERIALS AND METHODS

### Study species and measurements

The experiment was conducted in a nest-box population of blue tits near Antwerp, Belgium (Peerdsbos 51°16′N, 4°29′E) from April to May 2016. Nest boxes ($n = 131$) were checked twice per week for nest building, egg laying and incubation. Nests were monitored daily for hatching from the expected hatch date onward. The first day of hatching was defined as day 0. Parents were caught on day 6 for individual measurements and were given a plastic leg band with an integrated PIT tag (which is a standard procedure in our field site and aids the individual recognition by placing the PIT tag on different sides for the individuals of a pair) (*Iserbyt et al., 2018*). Nestlings were given a unique metal ring on day 6 to allow individual recognition.

### Brood size manipulation

Each nest was subjected to a control (original brood size), reduced (minus three nestlings) and enlarged (plus three nestlings) treatment each on a separate day, that is, post hatching day 8, 9, 10 (*Parejo & Danchin, 2006*; *García-Navas & Sanz, 2010*). This design was chosen to control for potential confounding effects of brood age and date. The specific observation period was selected to make sure that females do not brood the young anymore (females brood nestlings 6–7 days when hatched (*Perrins, 1979*)), given confounding effects of task specialization on alternation (*Iserbyt et al., 2017*). The three nests that were chosen for the triad structure were matched for hatching date (no difference allowed), mean nestling mass (maximum difference of 1.5 g) and brood size (maximum difference of two nestlings) on day 7. When only two nests were available on the same hatch date, dyads were made considering the same selection criteria for brood size and mean nestling mass. Nests with less than seven nestlings were not used for this experiment.

Each treatment was set up in the morning and terminated in the late afternoon (8 AM–6 PM). At the start of the experiment, an infrared nest-box camera (420TVL; Pakatak PAK-MIR5, Essex, UK) was placed facing downward to the nest to record

parental nest visits and prey sizes (*Lucass et al., 2015*). Furthermore, all nestlings were weighed individually in the morning at the start of the brood size manipulations. The three nestlings that were used for swapping between nests belonged to the intermediate positions within the weight hierarchy and did not differ more than one gram relative to the other selected nestlings from the other nests in the triad. Those three nestlings were taken from the nest that got the reduced treatment and were transported to the enlarged treatment nest in a cloth bag. In the late afternoon when the brood size manipulation was terminated, all nestlings were weighed again and nestlings were returned to their original nest.

## Statistical analyses

Parental nest visits and prey sizes were scored from the recordings during the afternoon (starting at 4 PM), to allow the parents to adjust to their new brood size that was manipulated in the morning. We performed a pilot analysis, in order to determine the minimum number of visits that are required to provide a reliable visit rate estimate. This analysis was based on eight control videos and showed that the average visit rate remained fairly consistent beyond 10 visits. More specifically, doubling the number of visits from 10 to 20 visits provided very similar visit rates (Pearson correlation: $r = 0.84$; $P < 0.001$). Therefore, the videos were analyzed until each sex had at least 10 visits or the analysis was terminated after 2 hours when still one of the parents had not visited for 10 times. The mean time window for an analyzed video recording until a parent had a minimum of 10 visits, was $51.5 \pm 3.5$ min (mean $\pm$ SE, $n = 55$), which is likely to provide accurate estimates for individual visit rates (*Pagani-Núñez & Senar, 2013*; *Lendvai et al., 2015*). The size of the bird beak was used as a reference to estimate prey size: 1 = small (<1 beak length), 2 = medium (1–3 beak lengths), 3 = large (>3 beak lengths) (sensu *Kölliker et al., 1998*; *Lucass et al., 2015*). All videos were analyzed by M. Griffioen using the ObserverXT program (version 10.5.572, 2011, Noldus Information Technology, Wageningen, The Netherlands). Males and females were distinguished by their PIT tag (different sides or color) or by their head pattern (males have dark stripes on their head which are visible on the infra-red images due to the gray scale of the videos). Only nests at which both parents visited were used in the analyses because we are interested in the cooperation of the parents ($n = 21$ out of 23). Due to failures of video recordings we obtained 18 observations of control situations, 17 of reduced and 20 of enlarged. Field work was carried out under the license from the Ethical Committee for animals (ECD) of the University of Antwerp (license number: 2015-85).

Visit rate was calculated as visits per hour for each parent separately from the video recordings. The alternation was calculated as a pair score via $F/(t-1)$ with $F$ being the number of visits that were alternated and $t$ the total number of visits (as in *Bebbington & Hatchwell, 2016*). The proportions of prey sizes (small, medium and large) were calculated separately per treatment for each parent. Mixed effect models were used to analyze how the different parental provisioning estimates varied with our brood size manipulation. The first model (linear mixed model with Gaussian distribution) investigated whether variation in parental visit rate (log transformed) could be explained by experimental treatment (control/reduced/enlarged), sex and the interaction between

treatment and sex. Brood age was included as a covariate. Parent ID nested in nest ID was included as random effect to avoid pseudo-replication. The second model (generalized mixed model with binomial distribution) analyzed the effect of treatment on parental alternation. Treatment was included as fixed factor and brood age as a covariate. Only nest ID was included as a random factor because alternation is measured here as a couple parameter. To further explore the effects of alternation we investigated whether visit rate increased with a higher level of alternation (see *Johnstone et al., 2014*; *Bebbington & Hatchwell, 2016*, but see *Iserbyt et al., 2017*). A linear mixed model (Gaussian distribution) was used in which variation in the sum of male and female visit rates (total provisioning, log transformed) was explained by alternation score, treatment and their interaction. Nest ID was added to the model as random effect. The fourth and fifth model (generalized mixed models with binomial distribution) investigated the effect of the brood size manipulation on prey sizes. The fraction of small and medium prey sizes were used as dependent variables in these models. Treatment, sex and the interaction between treatment and sex were used as fixed factors and brood age as covariate. The random factor was parent ID nested in nest ID. The large-sized prey were not analyzed because the proportions were too low and zero-inflated. To investigate whether and how alternation affects offspring growth, a linear mixed model (Gaussian distribution) was fitted with the change in nestling mass as response variable. Nestling mass was taken each day at the start and the end of the brood size manipulation, resulting in a mean weight gain for each nest (only resident nestlings). These weight gains were corrected for the variation in time period between both measurements so we acquired an average nestling mass gain per hour for each nest. The total provision, alternation scores, treatment and the interaction between treatment and alternation were included in the model as a fixed effects and brood age as covariate. One outlier in mass gain was removed from the analyses to get the model residuals normally distributed.

All mixed models were run using the package lme4 (*Bates et al., 2015*) and lmerTest (*Kuznetsova, Brockhoff & Christensen, 2017*) in R studio (version 1.1.423 and R version 3.4.3, *R Core Team, 2017*). To investigate the significance of the fixed factors we performed backward stepwise elimination with a critical $\alpha$ level of 0.05. In two models the response variables "visit rate" and "total provisioning" were log transformed to meet the normality of assumption of the model residuals, tested with a Shapiro normality test and visual inspection of the residuals.

## RESULTS

### Parental feeding rates

There was no significant interaction effect between treatment and sex on parental visit rates ($F_{2, 69.5} = 0.30$, $P = 0.739$; Fig. 1). Parents changed their visit rate according to experimentally manipulated brood sizes ($F_{2, 70.3} = 18.7$, $P < 0.0001$). Specifically, parents decreased their visit rate from control (unmanipulated) to reduced brood sizes (differences of LSmeans: $t = 2.73$, d$f = 68.9$, $P = 0.008$), and increased from control to enlarged brood sizes (differences of LSmeans: $t = -3.22$, d$f = 67.5$, $P = 0.002$). Furthermore, females had an overall lower visit rate than males ($F_{1, 20.4} = 9.75$, $P = 0.005$; Fig. 1).

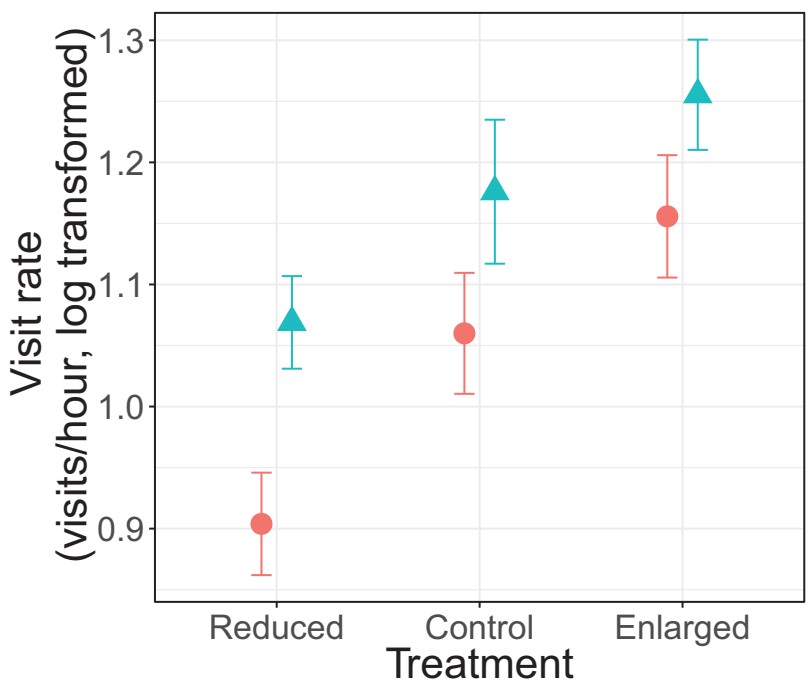

**Figure 1 Parental feeding rates (visits/hour, log transformed) for each treatment (reduced ($n = 17$), control ($n = 18$) and enlarged ($n = 20$) brood size) and sex (blue triangles: males, red circles: females).** Symbols represent means ± SE.

Brood age also had a significant effect ($F_{2, 70.2} = 5.72$, $P = 0.005$). The visit rate was higher when nestlings were 8 days old when compared to 9 days old nestlings (differences of LSmeans: $t = 3.34$, d$f = 69.5$, $P = 0.0014$) but not with 10 day old nestlings (differences of LSmeans: $t = 1.60$, d$f = 69.0$, $P = 0.113$).

## Parental alternation

The alternation of feeding visits did not change as a function of our brood size manipulation ($\chi^2 = 1.15$, d$f = 2$, $P = 0.564$; see Fig. 2). Brood age also did not significantly affect alternation ($\chi^2 = 4.68$, d$f = 2$, $P = 0.096$). The analyses of the model with total parental visit rates revealed no interaction effect of treatment with alternation ($F_{2, 38.7} = 0.192$, $P = 0.826$; see Fig. 3) and as well no overall relationship with alternation scores ($F_{1, 48.7} = 0.027$, $P = 0.870$) or brood age ($F_{2, 34.1} = 3.07$, $P = 0.059$).

## Prey sizes

The effect of the brood size manipulation on the proportion of the small prey size did not differ between the sexes ($\chi^2 = 3.65$, d$f = 2$, $P = 0.161$; see Fig. 4). However, the brood size manipulation had a significant effect (treatment: $\chi^2 = 8.54$, d$f = 2$, $P = 0.014$) with birds bringing a higher proportion of small prey items in the control treatment when compared to the enlarged treatment (post hoc Tukey: enlarged—control $Z = -2.91$, $P = 0.010$), whilst there was no difference in proportion of small prey items when control was compared to the reduced treatment (post hoc Tukey: reduced—control $Z = -1.47$, $P = 0.304$). Overall males brought more small prey items than females
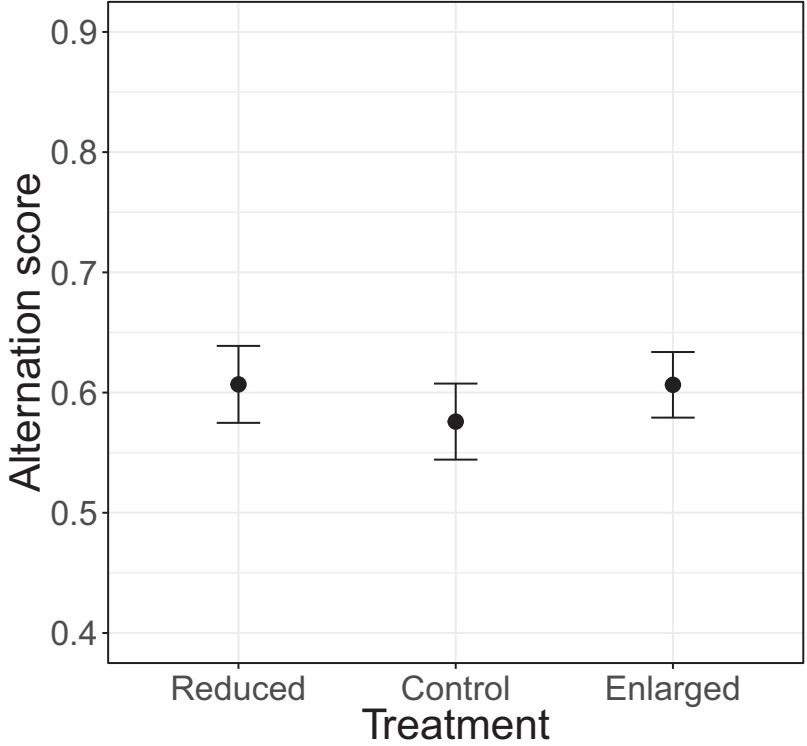

**Figure 2 The alternation levels (number of alternated visits/total number of visits − 1) for each of the treatments (reduced *n* = 17, control *n* = 18, enlarged *n* = 20).** Symbols represent means ± SE.

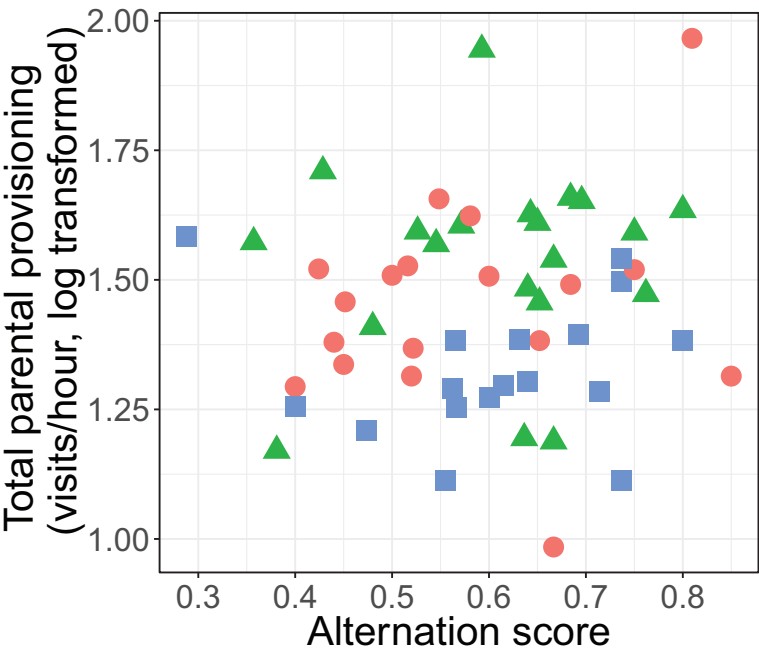

**Figure 3 Total parental provisioning rates (visits/hour, log transformed) per nest in relation to alternation score (number of alternated visits/total number of visits − 1).** The symbols indicate the different treatments (red circle: control brood size (*n* = 18), green triangles: enlarged (*n* = 20), and blue squares: reduced (*n* = 17)).

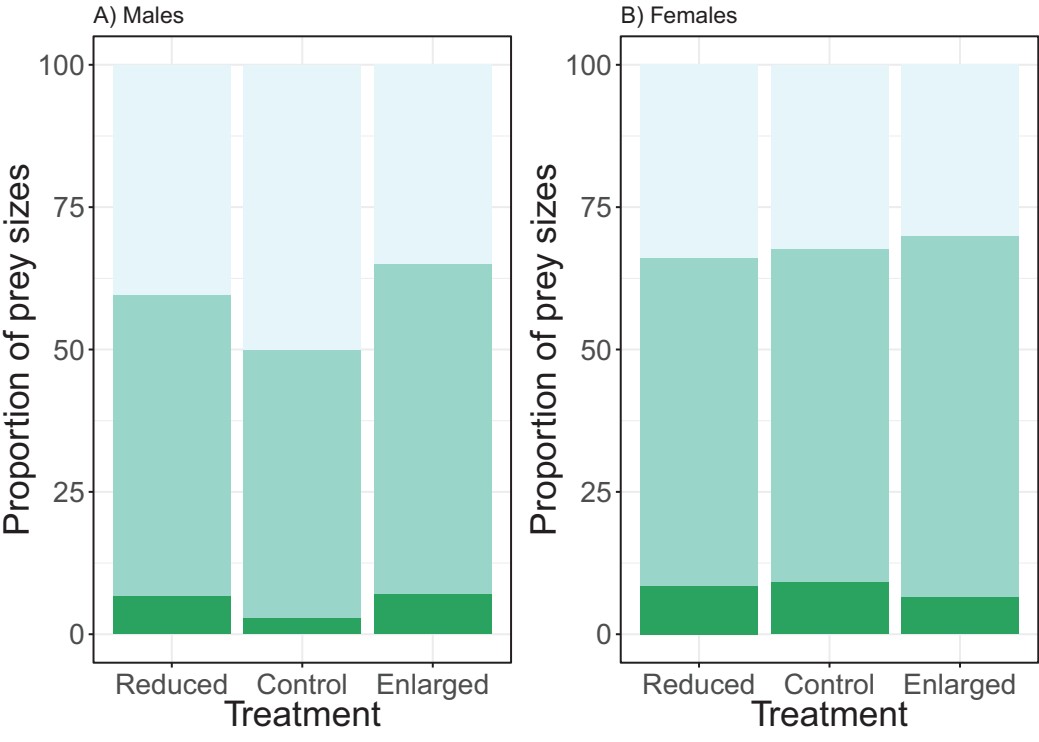

**Figure 4** **The proportions of prey sizes of (A) males and (B) females for the different treatments (reduced ($n = 17$), control ($n = 18$) and enlarged ($n = 20$) brood size).** The dark green bars (lowest bars) indicate the proportions of large prey items, medium green (middle bars) the proportions of medium-sized prey and the light green (top bars) the proportions of small prey. Symbols represent means ± SE.

(post hoc Tukey: $Z = 2.73$, $P = 0.006$). Brood age did not affect the proportion of small prey brought to the nest ($\chi^2 = 4.39$, d$f = 2$, $P = 0.116$).

The analyses of the proportion of medium prey sizes revealed similar effects. There was no significant interaction effect of treatment and sex ($\chi^2 = 0.78$, d$f = 2$, $P = 0.677$). Brood size manipulation had a significant effect on the proportion of medium prey (treatment: $\chi^2 = 7.43$, d$f = 2$, $P = 0.024$), with the proportion of medium prey sizes being higher in the enlarged than in the control brood sizes and no change between the control and reduced (post hoc Tukey: enlarged—control $Z = 2.64$, $P = 0.023$; reduced—control $Z = 0.74$, $P = 0.737$). There was a significant sex effect, females brought more medium prey items compared to males (sex: $\chi^2 = 3.97$, d$f = 2$, $P = 0.046$; see Fig. 4). Brood age did not affect the proportion of medium prey items ($\chi^2 = 1.72$, d$f = 2$, P $= 0.422$).

## Offspring mass gain

There was no significant overall effect of alternation ($F_{1, 45.3} = 0.135$, $P = 0.715$), treatment ($F_{2, 35.2} = 2.53$, $P = 0.094$), total provisioning ($F_{1, 46.1} = 0.05$, $P = 0.823$) and brood age ($F_{1, 33.9} = 0.0003$, $P = 0.989$) on nestling mass gain. Neither did the effect of alternation on the mass gain of nestlings differ between treatments ($F_{2, 43.2} = 1.66$, $P = 0.202$; see Fig. 5).

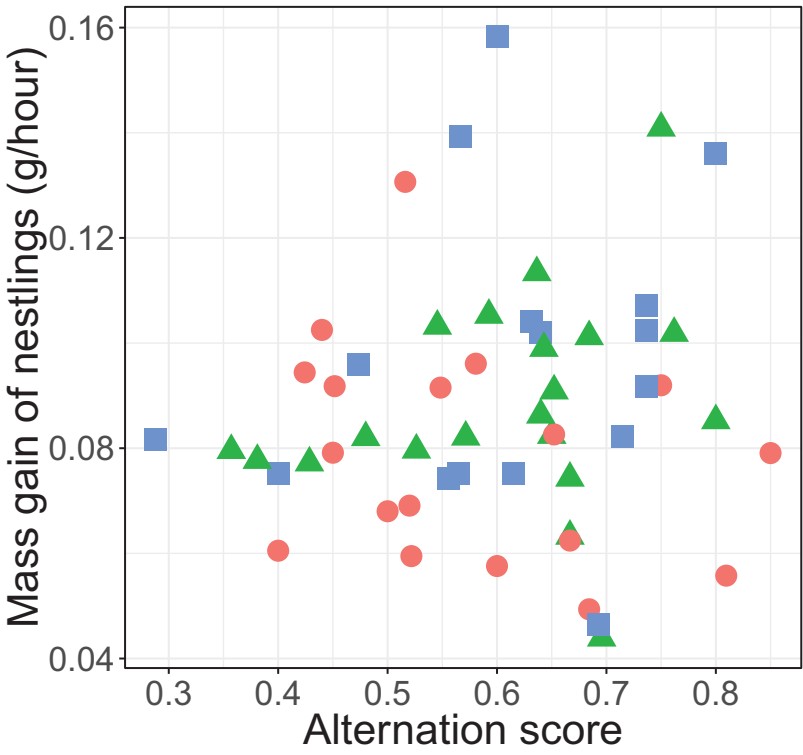

**Figure 5 Mass gain of the nestlings (gram per hour) and alternation level of their parents.** Alternation was calculated as number of visits alternated divided by total number of visits minus one. The symbols represent the reduced (blue squares; $n = 16$ (one outlier removed, see statistical analyses)), control (red circles; $n = 18$) and enlarged (green triangles; $n = 20$) treatment.

## DISCUSSION

Our study shows that blue tit parents adjust their visit rates in response to experimentally manipulated brood sizes. Couples nevertheless maintained similar levels of cooperation as their alternation of feeding visits did not change in function of the brood size manipulations. We discuss the implications of our results for the resolution of parental conflict and conditional cooperation theory, as well as the functional consequences for offspring development.

### Effects of brood size manipulation

We hypothesized that parents when faced with extra nestlings have to increase their visit rates to meet the enhanced offspring demand, which likely reinforces sexual conflict over parental care. Parents may negotiate and resolve this conflict by increasing the level of alternated visits, because conditional cooperation and therewith alternation should enhance each other's willingness to invest (*Johnstone et al., 2014*). The blue tit parents in our study indeed increased their visit rates when confronted with three extra nestling, which is in line with observations in several other species (house wrens: *Bowers et al., 2014*; great tits: *Hinde & Kilner, 2007*; blue tits: *Nur, 1984*; *Parejo & Danchin, 2006*; *García-Navas & Sanz, 2010*). However, we did not observe an increase of

the alternation levels. Therefore, the extra investment and thus the probably reinforced conflict seems not to be mediated by alternation. Furthermore, parents additionally increased their investment when confronted with extra nestlings by increasing the number of medium-sized prey while lowering the proportion of small prey items. This contrasts previous studies that bringing smaller prey items could possibly minimize the search time for prey facilitating an increase of visit rates (*Nour et al., 1998*; *Grieco, 2002*). However, both sexes show the same pattern in prey size which makes it unlikely that parents cheat in their investment via prey sizes. Yet, females brought overall more medium-sized prey items than males suggesting a sexual difference in investment. However, this difference is likely mitigated by the lower visit rates of females, relative to males.

For the reduced nests we hypothesized that parents would decrease their alternation due to lower conflict about investment caused by lower brood demand. Indeed, the blue tit parents in our study did lower their visit rates as predicted. However, again their alternation level was not changed. Furthermore, prey sizes were not altered in the reduced manipulations by the parents. When brood demand and consequently visit rate is reduced, more time becomes available as potential search time for big profitable prey (*Nour et al., 1998*; *Grieco, 2002*). However, both males and females did not elevate the small or medium prey sizes. The prey size that they brought to the nest might have been optimal in terms of foraging efficiency which is determined by the search time and energetic value of the prey (*Naef-Daenzer & Keller, 1999*; *Naef-Daenzer, Naef-Daenzer & Nager, 2000*).

A meta-analysis revealed that partial compensation is the general parental response in mate handicapping or mate removal studies (*Harrison et al., 2009*). However, manipulation of one or both parents can also result in no response (*Schwagmeyer, Mock & Parker, 2002*; *Santema et al., 2017*) and even a matching response (*Hinde, 2006*; *Meade et al., 2011*). Such variation in parental responses may in part stem from multitude of applied methods and may even be observed within the same study. For example in the same blue tit population which is used here, a sex-specific partial compensation was observed in response to a temporal mate removal experiment, but a clear matched response followed upon the reunion of the pair (*Iserbyt et al., 2019*). Interestingly, we here found that male and female parents responded in similar, matched ways to the brood size manipulation (*García-Navas & Sanz, 2010*), which allows limited fluctuations in within-pair alternation levels (see further, *Bebbington & Hatchwell, 2016*).

## Effects on offspring development

If alternation represents a form of conditional cooperation, in which parents enhance each other's willingness to invest, it should benefit the offspring development. Here, we did not find a relationship between total provisioning and alternation which suggests that there is no positive effect for nestlings (see also *Iserbyt et al., 2017*, but see *Bebbington & Hatchwell, 2016*). Consequently, there was no evidence that higher levels of alternation resulted in increased nestling mass gain. However, the duration of the manipulation

might have been too short to detect any differences in mass gain. Unfortunately, studies investigating the effects of conditional cooperation on offspring development are still limited and thus far have shown contradicting results (benefit: increased brood success, *Bebbington & Hatchwell, 2016*; no effect: offspring development and physiological state; *Bebbington & Hatchwell, 2016*; *Iserbyt et al., 2017*). That renders it difficult to draw definite conclusions about the adaptive significance and fitness consequences of alternation. We suggest that future research should not only incorporate measures of offspring development, but also estimates of parental fitness. In particular since conditional cooperation should be costly for the caring parents, or at least for the parent with the lowest physiological condition.

## CONCLUSIONS

Alternation of parental nest visits is thought to be the outcome of a behavioral mechanism in which both parents co-adjust their visit rates, thereby avoiding exploitation by their partner (*Johnstone et al., 2014*). However, in our study we did not find such evidence that alternation and visit rates (total visit rates) were related (*Iserbyt et al., 2017*, but see *Bebbington & Hatchwell, 2016*). Furthermore, alternation remained rather constant regardless of experimentally induced changes in parental visit rates. Consequently, parents might have had a fixed agreement about the level of alternation (*Iserbyt et al., 2019*). They might use this to control for any free riding of their partner and it might have already been established during earlier phases of the reproductive cycle. Thus independent of the environmental conditions and brood demand, the proportion of alternated visits should be kept relatively constant within pairs. This, however, does not necessarily require that all visits should be alternated, especially when it might be costly to monitor each other. A specific fraction of monitoring the partner's feeding behavior might provide a sufficient estimate of its investment and could serve as a signal to avoid substantial exploitation. This might imply that the level of alternation should thus mainly vary among but not within couples. The variation in alternation among couples could relate to quality differences between parents or the compatibility of pairs (*Ihle, Kempenaers & Forstmeier, 2015*). However, this remains as yet speculative as we do not know if and when such an agreement is established. In this context, further research needs to specifically consider how differences in quality of parents within a pair affect the alternation levels, as they have the potential to lower the levels of care toward the parent of lower quality. Furthermore, environmental factors that can shape the distribution of provisioning visits via prey abundance and patchy distributions in the pairs' environment should be investigated to understand the role of environmental effects on alternation.

## ACKNOWLEDGEMENTS

We thank B. Briesen, P. Scheys and N. Fresneau for assistance in the field, J.M. Baert and J.L. Savage for statistical advice and W.F.M. van Andel for general support and discussion. A. Lendvai and an anonymous reviewer for providing constructive comments, which considerably helped to improve the clarity of this manuscript.

### Funding

This work was supported by predoctoral and postdoctoral grants from the Fonds Wetenschappelijk Onderzoek – Vlaanderen (FWO) (Project ID: 1143817N to Maaike Griffioen; 1517815N and 12I1916N to Arne Iserbyt). The funders had no role in study design, data collection and analysis, decision to publish, or preparation of the manuscript.

### Grant Disclosures

The following grant information was disclosed by the authors:
Fonds Wetenschappelijk Onderzoek – Vlaanderen (FWO) (Project ID: 1143817N to Maaike Griffioen; 1517815N and 12I1916N to Arne Iserbyt).

### Competing Interests

The authors declare that they have no competing interests.

### Author Contributions

- Maaike Griffioen conceived and designed the experiments, performed the experiments, analyzed the data, contributed reagents/materials/analysis tools, prepared figures and/or tables, authored or reviewed drafts of the paper, approved the final draft.
- Wendt Müller conceived and designed the experiments, authored or reviewed drafts of the paper, approved the final draft.
- Arne Iserbyt conceived and designed the experiments, performed the experiments, contributed reagents/materials/analysis tools, authored or reviewed drafts of the paper, approved the final draft.

### Animal Ethics

The following information was supplied relating to ethical approvals (i.e., approving body and any reference numbers):

Field work was carried out under the license from the Ethical Committee for animals (ECD) of the University of Antwerp (license number: 2015-85).

### Data Availability

The data that was used in the statistical models is available as a Supplemental File.

### Supplemental Information

Supplemental information for this article can be found online at http://dx.doi.org/10.7717/peerj.6826#supplemental-information.

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
