# Peer review of "A fixed agreement—consequences of brood size manipulation on alternation in blue tits"

_PeerJ, doi:10.7717/peerj.6826_

## Round 0.1 · original submission · Major Revisions

Both referees agreed that this is an interesting manuscript; however, they also had several suggestions as to improve it. In particular, both referees questioned the relevance of the statistical analyses (altough they do not think that using a more appropriate modelling approach would change the results). This has to be fixed in the revised version. They also suggested that the methods should be better explained to allow study's reproducibility. I am looking forward to receiving the revised version.

Reviewer 1 ·

Basic reporting

The article is competently written, with sufficient references. Some rewording could be useful in places to explain the rationale behind some of the predictions, but overall the hypotheses and results align well. Revisions are needed in the presentation of the figures - details below. Data underlying the article is provided.

Presentation:
Figures 1 & 2: Axes labels and plot symbols need to be much larger, and units are required for prey size. Figure 2: Units or explanation (in legend) required for deviation from expected alternation score.
Figures 3 & 4: Again, axes labels need to be much larger to be easily readable. Different symbols very difficult to distinguish – these also need to be much larger. Either the figure legend or the axes need to make it clearer what the units are for the deviation from expected alternation score.

Other comments:
47: ‘Parents were expected to visit more and more consistently alternate their visits’ would be clearer wording here.
49: Not clear what ‘less relevant’ means here.
60: It’s odd for the citation here to be a chapter of a book, and then the entire book. Also, the 90% figure for biparental birds is taken directly from a single paper (Cockburn 2006 PRSB), so that paper should be cited here instead of the chapter. Furthermore, the 90% figure includes cooperative species where both parents care, which are not ‘biparental’ in the relevant sense for this paper of only having two carers (a more accurate number would be 81%).
132-4: Move this sentence down – it mentions video recordings before any description of the recording methods.
147-8: This sentence is part of the statistical methods and is not needed here.

Experimental design

The article fits within the journal aims and scope. Research gap identified, but rationale for research question and approach could be improved. No concerns about ethics or technical approach taken. Methods need some additional details for reproducibility.

Comments about research question:
50-1: Why would alternation lead to more weight gain? I would expect visit rate and prey size to be all that matters here. Unless the suggestion is that parents of offspring that gain more weight should both visit more and be better-coordinated… but in that case why not just predict that visit rate and alternation should be correlated with each other?
95: What does a ‘stable parental strategy’ mean?
99: Is a brood size manipulation changing ‘environmental conditions’?
109-12: This explanation is clearer than the one in the abstract, however the causality is still unclear to me. The brood size manipulation is intended to manipulate visit rates (which may then alter the level of alternation), but here it reads as if it’s alternation driving visit rates. Is this meant to be a prediction that in general one should expect pairs that alternate more to have higher visit rates, as they have lower levels of sexual conflict?

Comments about reproducibility:
136: Source company of camera should be added.
138-9: Some information about how these prey size scores were assigned should be included in the text rather than just cited, as this is a key variable in the following analysis.
145-6: With L125 this suggests that all your visit data are from between 4pm and 5pm, but with L144 this suggests you analysed up to two hours from 4pm. Which is accurate?
162: To clarify (since you stopped at 10 visits from either parent), are the visit rates for each calculated over different observation windows?

Validity of the findings

Conclusions clearly linked to original question, discussion and speculation reasonable. I have a number of concerns about the statistical approach that might influence the results. The most important of these are (1) the comparison of observed values to the mean of randomised values, without accounting for the variance of those randomised values; (2) analysis of the ordinal prey size variable as if it were continuous, (3) some missing details on how the statistics were implemented.

Comments about statistics:
143: How are you judging whether these constitute ‘very similar’ visit rates? The Pearson test just tells you that the values are correlated (which will obviously be the case) – did you have some threshold r value you used to choose the number of visits to analyse?
170-3: The variance of the randomisations is also important here – the absolute difference between the observed alternation and the mean of the randomisations does not tell you much without knowing the spread of the randomised data.
173-4: I don’t think it is appropriate to calculate a mean from ordinal data in this way – only the median and mode are defined. Depending on the distribution of S/M/L items, ‘fraction of small items’ or similar might also be suitable.
180: Was there a reason for not including visit rate as a predictor of alternation? This would seem relevant to your question.
181: Do you know whether alternation was related to prey sizes? This is mentioned as being important above but I couldn’t see any analysis of it.
185: A cleaner analysis here would be to have a model with both male and female visit rates as the response variables, and your existing factors as predictors.
195: This package only adds p-values to models that are run by other packages (e.g. lme4)
196: Not clear what you mean by this: you don’t present minimal models in the results, just the significance levels of each fixed effect.
198: What is ‘total provisioning’? Why is this in the model as well as visit rate, when they are presumably highly correlated?
212-4: I don’t follow the statistics used here. Is this a sign test (or similar paired test) comparing the observed values to the means of the randomised values?
215-6: I don’t think the deviation is a meaningful value without an estimate of the variance.
229: I don’t understand the rationale of any potential effect of alternation on mass gain.

Comments on discussion points:
242: Why does this change need to occur via alternating more, rather than just directly as a response to increased demand?
249: I don’t think Johnstone et al. suggests that there is really a ‘feedback loop’ – just that one parent increasing investment encourages the other to do so immediately and to the same degree, rather than the (potentially delayed) incomplete compensation predicted by other models.

Additional comments

Overall, I think this is a competent piece of work that mainly requires 'polishing' to be suitable for publication. While I made a relatively large number of comments above, many of these are minor problems of detail, clarity, or presentation. The more substantial issues should at most require relatively simple reanalyses, so I encourage the authors to persist in improving the article.

·

Basic reporting

This is an interesting study that investigates the effects of experimental manipulation of provisioning behavior (via brood size manipulation) on parental coordination (i.e. alternation of feeding visits). The manuscript is generally well written, but improvements are necessary at some places (see below).

Experimental design

There is no explanation for the rationale of the experimental design chosen. Why were brood sizes manipulated at an advanced stage of the nestling period (days 8-10)? What was the goal of the repeated-measure design, especially, that no within-nest analyses were performed? Given this short-time manipulation, it is hard to expect any differences in the nestling mass, and actually the study did not find a treatment effect here.

The description of the methods is sometimes confusing or not detailed enough.

Validity of the findings

I think the statistical analyses could be improved - although I don't think that it would alter the conclusions.

Additional comments

1. I'm not 100% convinced by the conceptual framework of the study. First the description of the theoretical background is a bit superficial. McNamara et al. 1999 were not the first to suggest a bargaining process reciprocally settling mate efforts at limiting values (see e.g. Chase 1980). Next, Johnstone and Hinde (2006) did not simply predict that "parents should invest below the most efficient level of care". But most importantly, one can argue that while higher alternation may lead to higher overall feeding rate, the reverse should not necessarily be the case, i.e. increasing the offspring need would not lead to higher alternation (and as the study finds, it does not). Put it simply, if the chicks are hungry, the parents are busy searching for food and as soon as they found a food item, they should bring it to the offspring. If they are waiting for their parents to feed first, that would limit their provisioning rate. Actually, one may argue that alternation should be higher when the offspring need is decreased, because in that case parents can afford bargaining about provisioning.

2. I also think that the way alternation is calculated is too restrictive, because it only considers immediate reciprocity, and disregards any other temporal pattern in the data. To illustrate this point, imagine a nest, where males (M) and females feed their young in this sequence:

MFMFMFMF - this is 1.0, i.e. maximal alternation score.

Now consider

MMFFMMFF - which is also alternated, but follows the rule "you feed twice, then I feed twice", but the alternation score is merely 0.42

And now consider

MMMFMMMF - which also yields an alternation score of 0.42, despite being much less egalitarian than the sequence MMFFMMFF.

Considering temporal autocorrelation in the data may be a more sensitive way to analyze the data.

3. The structure of the Methods should be improved. For example, the description of the brood size manipulation experimental design is hard to understand (lines 125-131). Only from line 178 we figure it out that each nest was subjected to all treatments. This should be made much more clear.

In line 133, you mention "during the video recording", but this is the first time we learn that video recordings were used at all. The description of the video analyses needs more details. For example, who analyzed the videos? Was that the same person? Was that person blind to the treatments? How was the repeatability of measurements (e.g. prey size)? How did you determine the sex of the parents using IR cameras?! The data file contains a variable PIT tag number, but there is no mention of this in the text. When were the parents captured? What were the PIT tags used for? Etc.

In line 147, you say that alternation scores were calculated, but we don't understand how until we reach the next section - this is confusing.

4. Statistical analyses. Since visit rates are actually counts, with usually a skewed distribution, you could consider analyzing this variable using a Poisson model. The alternation scores are actually proportion variables, so these should be analyzed in a binomial model. The third model should be a multinomial model (only 3 categories). Analyzing these variables in a Gaussian model is not correct.

Brood age should be controlled for in the models.

5. Sample sizes
line 117 and throughout: the sample size should be denoted with a lower case 'n'. 'N' is usually the symbol of the population size, the the sample size. Sample sizes should be explicitly included in the figures. It is not explained why sample sizes differ between Fig.3 (I counted 17, 18 and 20 circles, triangles and squares respectively), and Fig 4 (18, 19, 16)?

---

## Round 0.2 · Minor Revisions

Both referees found that the manuscript was improved, but still had a number of suggestions that should be taken into account in a final revision.

Reviewer 1 ·

Basic reporting

1. The manuscript is structured well, but is still in need of revisions to ensure it conveys its key points clearly, particularly in the introduction and discussion. Specific points I think are in need of clarification:
L42 – Not immediately clear what ‘This’ is referring to.
L43 – Not clear what ‘alternation’ is.
L62 – While it is obvious what you mean, this sentence reads as if birds are not animals.
L78 – Not clear what ‘efficient’ means in this context (and this is important for your argument later).
L88 – This sentence needs rewording to make it clear this is an extension of the previous sentence attempting to define conditional cooperation in this particular context.
L102 – Not clear what ‘whether alternation represents a parental strategy’ means. From the context it seems like you mean a specific level of alternation, but this needs to be clear.
L107 – I would add in ‘through increasing individual costs of parents’ to the end of the line here, to ensure the point is obvious later in the paragraph.
L115 – I don’t know what this last sentence is trying to convey.
L281-3 – I don’t follow the argument of this sentence at all – wouldn’t difficulty tracking prey size make it easier for parents to cheat?
L297 – Was perfect alternation expected? If so, why? If not, what is this section trying to convey?
L325 – While looking at fitness more broadly is a reasonable suggestion, in the context of this study the point could be made more clearly. The key take-home of conditional cooperation (as discussed in the Johnstone et al. model cited above) is that the outcome is more efficient than a negotiated or bidded outcome, i.e. closer to both parent and offspring optima. Presumably the argument here is that one should hence expect both (a) better outcomes for offspring and (b) higher costs for adults when cooperation between parents is stronger?
L329 – I think it would be clearer to say something like ‘thought to be a strategy by which parents can avoid exploitation by their partner during care, potentially leading to higher visit rates’.

2. Existing literature is reasonably well covered, however I would include at least one reference to an existing blue tit brood size manipulation study around L104-6 where the method is first mentioned, and again in the methods. In addition, I think the discussion would be improved by mention of some other studies, particularly those in which manipulations of parental effort have been attempted but failed to result in theoretically expected responses (perhaps around L275). This would provide context for the later interpretation of the results as suggesting a ‘fixed agreement’. For example, Schwagmeyer et al (2002) Behav Ecol 13:713-21 report an attempt to induce parents to alter their visit rates by handicapping partners, and found no appreciable response, suggesting a fixed ‘bid’ rather than a negotiation. Contrast could also be made to Santema et al. (2017) Anim. Behav. 123:117-27, another study on blue tits, in which a begging playback failed to elicit increased provisioning rates. Does this suggest brood size manipulations are more reliable at eliciting changes in feeding rate?

3. I suggest boosting the point sizes in figure 1 - the circles and triangles are too small to be distinguished from each other (and perhaps altering the size in figure 2, although all points are circles, to maintain consistency). In figure 5, 'gram' can be shortened to 'g'.

Experimental design

1. A few details are missing from the methods / results
L124 – Were the PIT tags part of a ring, injected, or attached some other way?
L200 – Presumably this was a LMM or GLMM as well?
L207 – Remove ‘linear’ here as you also presumably used lme4 for your GLMMs
L255 – Was there an effect of visit rate on nestling mass gain?

2. Some reordering would aid comprehension
L164-6 – This sentence belongs somewhere earlier, as currently the methods reads as if the cameras were set up, and then the following 10 visits were the ones analysed. In general this methods section should be carefully looked at to ensure it progresses in the most logical order.
L254-5 – I think it would make sense to present the overall effect of alternation before whether this effect differed between treatments.

Validity of the findings

On final revision, check speculation is clearly signposted.

Additional comments

Once all remaining reviewer comments are dealt with, I recommend the authors make a final pass through the article with a view to improving the readability of the text. There are numerous points at which sentences and/or topics either start or end abruptly, or are written as if related to or following from a previously unrelated point. Examples include L42, L115, L132, L140, L284, and elsewhere.

Minor corrections to wording:
L46 – remove ‘about’
L50-1 – ‘On the contrary’ -> ‘In contrast’
L148 – ‘that were actually biparental’ would be clearer as something like ‘at which both parents visited’.
L227 – ‘in function’ -> ‘as a function’
L238 – ‘amount’ -> ‘proportion’

·

Basic reporting

no comment

Experimental design

no comment

Validity of the findings

no comment

Additional comments

The authors satisfactorily addressed the comments raised by the reviewers and edited the manuscript accordingly. I think the manuscript improved a lot from the previous version. I think there are still a few outstanding questions and issues open for debate, but these do not affect the scientific integrity of the current manuscript. From a theoretical perspective, future articles may provide more specific hypotheses about when and how alternation is expected and future empirical studies may improve the way alternation is calculated. This paper may be a stimulation for these future works.

I only have a few minor issues that may need attention at the proofs stage:

1. Use consistent spacing following the journal's style recommendations. Currently, all possible combinations of using/not using space can be found sometimes just a few lines apart (e.g. 'n= 21' in line 149, '1=small' in line 154 and 'n = 55' in line 163). Sometimes an extra space is found (e.g. lines 227, 276). Please check the text throughout.
2. Fig. 5. The description of alternation in the legend does not match the description of the same variables in the previous figures. Please check. Also, the y-label should read 'mass gain of the nestlings' and not the 'nest'.

---

## Round 0.3 · accepted · Accept

This final version takes into account the points raised during the last round of review.

#